# eHealth and mHealth Development in Spain: Promise or Reality?

**DOI:** 10.3390/ijerph182413055

**Published:** 2021-12-10

**Authors:** Xosé Mahou, Bran Barral, Ángela Fernández, Ramón Bouzas-Lorenzo, Andrés Cernadas

**Affiliations:** 1Department of Political Science and Sociology, Faculty of Social Sciences and Communication, University of Vigo, 36005 Pontevedra, Spain; xmahou@uvigo.es (X.M.); angelafernandez.dasilva@usc.es (Á.F.); 2Department of Political Science and Sociology, Faculty of Political and Social Sciences, University of Santiago de Compostela, 15782 Santiago de Compostela, Spain; bran.barral.buceta@usc.es (B.B.); ramon.bouzas@usc.es (R.B.-L.)

**Keywords:** eHealth, mHealth, telemedicine, Spanish national health system, health policy, health equity, public policy, healthcare disparities

## Abstract

In the last decades, the use of Information and Communication Technologies (ICTs) has progressively spread to society and public administration. Health is one of the areas in which the use of ICTs has more intensively developed through what is now known as eHealth. That area has recently included mHealth. Spanish health system has stood out as one of the benchmarks of this technological revolution. The development of ICTs applied to health, especially since the outbreak of the pandemic caused by SARS Cov-2, has increased the range of health services delivered through smartphones and the development of subsequent specialized apps. Based on the data of a Survey on Use and Attitudes regarding eHealth in Spain, the aim of this research was to conduct a comparative analysis of the different eHealth and mHealth user profiles. The results show that the user profile of eHealth an mHealth services in Spain is not in a majority. Weaknesses are detected both in the knowledge and use of eHealth services among the general population and in the usability or development of their mobile version. Smartphones can be a democratizing vector, as for now, access to eHealth services is only available to wealthy people, widening inequality.

## 1. Introduction

### 1.1. How the National Health System Works

In the late 1970s, healthcare in Spain transitioned from a social insurance system that imitated those of central European countries such as Germany or Belgium to a national health system (NHS) that was universal in nature and funded exclusively through state taxes.

In tandem with this, the Constitution of 1978 decentralized the Spanish state into 17 Autonomous Communities (ACs). These were endowed with numerous competences, including management of healthcare services, while the state retained competence over non-domestic healthcare and general coordination of the healthcare system.

The new system materialized in 1986 with the Ley General de Sanidad (LGS, General Health Law), which expressed the political will to make the right to health protection effective, as recognized in Article 43 of the Constitution of 1978. The law established two fundamental principles: (1) that “All Spaniards and foreign citizens with residence established in the national territory have the right to health protection and medical attention” (Art. 1.2. LGS); and (2) that access to healthcare will occur in conditions of effective equality, and that this policy will be oriented towards overcoming social imbalances (Art. 3.2 and 3.3 LGS), according to the maxim of “equal access for equal need” [1].

### 1.2. eHealth

In recent years, Spain has been consolidating the application of new information and communication technologies (ICTs) to public administration. Many services understood as basic citizen rights have been digitalized through eGovernment programs. Healthcare was one of the first services to introduce ICTs.

Though reducing costs was the initial motivation, with ICTs administrative processes having been automated, attempts have been made to improve quality of service, to ensure professionals and users enjoy greater security, and to ensure that the system can be accessed using the internet [2]. However, investment has stagnated in Spain, in contrast with efforts of other European countries, to apply ICTs to public services and administration [3].

The integration of ICTs into health services and their use in diverse tasks related to healthcare and lifestyle management is known as digital health or eHealth. Recent research has introduced elements into the definition of eHealth that include the potential of ICTs for improving access to healthcare; increasing efficiency, efficacy, and the quality of clinical processes; and improving management for all actors involved [4,5,6].

Accessing the internet for information, along with new forms of communication, devices and applications that allow users greater control and monitoring of their health has been reported to contribute significantly to their empowerment [7]. These possibilities motivate users to take greater interest in their health [8], giving rise to ePatients: more proactive and informed individuals who seek to participate in decisions that affect their health [9].

Some eHealth services offered in Spain have been well received and are completely established, such as the use of electronic prescriptions, making appointments using the internet, or accessing Electronic Medical Records (EMRs) [10]. However, services such as telemedicine or activities related to digital imaging exhibit the need for greater investment and development to attain satisfactory levels of use. Several pilot programs [11,12,13] have offered positive preliminary results (reduced costs through tele-assistance or telecare) but lack sufficient continuity to become established.

In relation to eHealth, both the state as coordinator of the NHS, and the regional AC health services are adapting their web portals to facilitate access to services via mobile phones. To keep pace with the growing use of these devices, apps are being developed that allow users to interact with medical personnel or monitor activities or behaviors.

This provision of health services via smartphones is commonly known as mHealth. Some authors have indicated [14,15] that it could give a significant impulse to eHealth in general and telemedicine in particular [14,15].

### 1.3. mHealth

The term mHealth can be defined as “medical or public health practices that use mobile devices such as smartphones, personal digital assistants (PDAs), tablets and wearables” [15]. The use of these devices is linked to specific technical requirements and functionalities that include voice or text messaging, third, fourth, or fifth generation mobile telecommunications (3G, 4G, 5G), and Bluetooth or global positioning systems (GPS).

Health apps are on the rise. A study by Research2guidance reported more than 300,000 of them, mainly for the two great operating systems, Apple and Android [16]. Most of these are oriented to fitness or general wellbeing. The Institute for Healthcare Informatics [17] provides concurring data, indicating that the majority of such apps center on lifestyle (diet, physical exercise, stress, and way of life), followed by those that address specific aspects of health such as pregnancy, mental health, diabetes, or medication [18].

Similarly, a review of the relevant literature demonstrates that mHealth is mainly used in preventative health. The most frequently addressed topics are family planning, pregnancy, AIDS prevention, diagnosis, information about addictions or treatment, and follow-up of medications and pathologies [19,20]. Being a good tool [21,22], it is believed that mHealth could also have great potential for monitoring patients and chronic illnesses or conditions such as asthma, chronic pulmonary diseases, heart failure symptoms, blood sugar levels, and blood pressure. This approach is also highly useful for promoting health and gathering data to improve medical assistance and the system itself [23].

Though there is consensus about the convenience and ease of these technologies for professionals [24] and users, each of these actors has their own set of expectations and concerns. Healthcare professionals value mHealth very positively and are concerned with the stability of the programs, the potential usefulness of mHealth for their professional activity, and the security or reliability of data and internet connections [25,26]. Patients are concerned with usability and medical attention, the ongoing failure of devices to carry out many of the actions required for digital interaction to date, and app incompatibilities. They also express lack of confidence in security measures. Both groups clearly consider mHealth to be a good tool, but there are discrepancies about how specific utilities, such as access to digital medical records, should be developed [27].

The literature review revealed that research has centered mainly around studying—but very little on achieving—data usability, accessibility, and quality [28,29]. Significant interoperability issues derived from the incompatibility of gathered data are believed to persist [30]; the worldwide web is still considered insecure and lacking in investment [31]; institutional barriers have been detected [32]; and the lack of first-language information on the internet is accompanied by information reliability issues [33]. All these issues together create an important obstacle to the development of mHealth.

Studies seem to indicate that eHealth and mHealth do not generate social and health changes directly. Moreover, given that the technology incorporates forms of power, interests, and relationships between actors that are not always concordant [34], healthcare systems also require adaptation.

In this area, the outbreak of the SARS-CoV-2 virus that unleashed the global COVID-19 pandemic in 2020 has enlivened the debate about the potential of eHealth and mHealth as useful tools for controlling pandemics [35,36,37]. How users interact with healthcare professionals has changed significantly, as circumstances have accelerated the adoption of digital tools such as telemedicine and virtual care [32,38,39,40,41,42].

### 1.4. Objectives

This work provides an analysis of the use and recent evolution of eHealth in Spain, and mHealth in particular, based on the results of a survey involving a statistically significant sample of the population. The scientific literature was also reviewed, and relevant secondary sources were consulted from organisms that evaluate eHealth access and use, including the reports of the Observatorio Nacional de Tecnología y Sociedad de la Información (ONTSI, National Observatory on Technology and the Information Society) or the Instituto Nacional de Estadística (INE, National Statistics Institute), both official research Spanish institutions.

Motivated by the gap in the literature concerning the definition of an mHealth user profile [43,44,45], this work attempts to describe the levels of mHealth use and acceptance in Spain along with future perspectives. Specifically, the idea was to contrast users who opt to manage and consult health-related issues from a mobile device (mHealth) and those who access eHealth through other devices.

The study of individuals who use their mobile device (hereafter MobU) as an access point for eHealth services and those who use other devices, mainly computers, (hereafter WebU) to access the same services provides a reference for detecting similarities and differences in access and use of the online healthcare system and determining the degree of Health and mHealth penetration.

Accordingly, the results were configured and compared based on device use for each user profile (MobU, WebU) and ratings and opinions from both groups. In addition to the comparative aspect, the various survey question blocks made it possible to compile a panoramic picture of the state of eHealth services in 2018. It provides much more detail about uses and use profiles than what is offered in other secondary data sources.

## 2. Materials and Methods

Up-to-date information on the use of eHealth and mHealth services was gathered from the *Encuesta de Uso y Actitudes ante la eSalud en España* (Survey on Use and Attitudes regarding eHealth in Spain), hereafter referred to as the eHealth Survey.

### 2.1. Design of the eHealth Survey

This survey was conducted by means of telephone calls and Computer-Assisted Telephone Interviews (CATI). During the survey, opinions were requested about questions related to the accessibility for the main services offered on AC health web portals: scheduling a medical appointment, accessing digital medical records, managing electronic prescriptions, digital imaging, and telemedicine. Other items were included concerning use of devices, the perceived potential of new technologies, priorities for services, means of accessing internet, and perceived priorities about eHealth and mHealth services.

The average duration of the interview was 9.0 min, with a range of 5.7 to 12.3 min. The amplitude of the range was conditioned by filter questions, on the basis of which not all informants had to respond to the entire survey. The fieldwork was done from 24 May 2018 to 21 June 2018 throughout the entire Spanish territory, with the exception of the autonomous African cities of Ceuta and Melilla.

### 2.2. Population

A sample of 1695 adults legally residing in Spain were interviewed. Phone calls were randomly generated from the Infobel telephone directory at different times of day and days of the week. To guarantee adequate representativeness, sociodemographic profile features were included in the case selection process. Quotas were established for sex, age, and habitat, with a confidence level of 95% and a margin of error of 2.45 for the entire sample.

### 2.3. Analysis and Interpretation

The information gathered was stored in a codified data base according to the survey design and objectives. The preliminary results were corrected, standardized, and recodified into variables to facilitate statistical treatment. Weighting was applied with attention to the quotas mentioned earlier, to ensure representativeness at the national level. Finally, the data were thoroughly analyzed using SPSSTM and STATATM software packs. From these, general opinions were extracted and categorized for the set of informants, whose contribution to the research objectives was then assessed.

The analysis has been carried out with the aim of contrasting user profiles according to their use of health web portals on computer or mobile platforms. Health web portals have been the main or only gateway to eHealth services in Spain, and therefore, this centralized all access to them.

Based on this first selection of cases, a profile for each typology was created with the aim of determining the main differences and similarities between them. In addition, some indicators were built with the purpose of weighing and estimating the average use of services, thus defining the priorities of each profile. The operations carried out to perform these analyses are described along with the results in order to ease its understanding.

This treatment provided us with a description of the general situation of eHealth in Spain and made it possible to contrast the habits and expectations of two user profiles: MobU, indicating those who use a mobile device to access services, and WebU, indicating those who use computers or any other device for the same purpose.

## 3. Results

### 3.1. Dimension of Each Profile

As part of the WebU profile, the MobU contingent represented 24.4% of all those who interacted with eHealth services.

WebU comprised 49.91% of the Spanish population. This indicates that although the number of people who access eHealth services is considerable at nearly 50%, that use of eHealth is by no means generalized.

With this in mind, the two profiles as well as their use and expectations of eHealth and mHealth in Spain are described in the following sections.

### 3.2. Health Status and Frequency of Medical Visits

Mobile users (MobU) generally made greater use of in-person health services than web users (WebU). Although 59.4% of WebU went to the doctor between 0 and 3 times per year compared to 52.3% of MobU, the inverse situation was observed for the response category of 4 or more visits per year. In total, 47.7% of MobU made in-person visits compared to 40.6% of WebU.

Thus, MobU used healthcare services more frequently than WebU. Is this because the health status of MobU is worse or does it simply manifest their greater concern for their health? The following data on health status and the presence of chronic illnesses will help to clarify this question.

MobU and WebU self-assessed their health status almost identically (means and medians were extremely similar and hovered around 2, “Good”). MobU cases were higher at both ends of the distribution (especially in the “Very good” category but also in “Bad” or “Very bad”), but not notably higher than WebU (less than 2%).

For chronicity, the similarity continued: 35.7% of WebU declared a chronic condition compared to 33.3% of MobU.

### 3.3. Use of Internet (General and Specific to the Medical Context)

When asked about how they connected to the internet, MobU indicated that their main device was the mobile phone: 85.6% used it every day and 93.2% used it frequently. After that, and to a much lesser degree, other devices named included desktop computers, laptops, and tablets, which 20% reported using daily and 30–45% reported using frequently.

The WebU group also used their mobile phones as their main tool for accessing the internet, but the numbers were lower than MobU; in fact, about 10% less of WebU reported frequent use (83.8%). Desktops and laptops filled the gap in similar numbers, though slightly higher for desktops: over 30% indicated daily use and nearly 50% reported frequent use.

Use of tablets to access the internet was almost identical between profiles, with nearly 20% reporting daily use and slightly more than 30% reporting frequent use.

Finally, 34% of both user profiles accessed the internet with their television to some degree, with over 10% indicating that they go online daily with this device. These figures differentiate the access profiles of the two groups through infrastructural or technological means.

When asked about the physical space or location from which they usually connected to the internet, the MobU group was less restricted to a specific location. Nearly one-third of MobU connected from anywhere, and nowhere in particular, compared to 18% of WebU. In that group, domestic/home spaces emerged as clearly predominant places of internet connection in 70% of the cases, compared to 62.1% of MobU.

Finally, the third most relevant option of connecting from the workplace was reported by 10.6% of WebU and 7.6% of MobU.

Despite the differences, connection from home remained predominant. This may be associated with consumption and free-time habits, despite the significant numbers for “free” access (no preference for a specific location), especially among people who access the internet from their mobile phones.

For general online activities, both profiles displayed the same priorities and order in frequency of use, but with clear differences in the intensity of task completion.

As the following graph illustrates (Figure 1), MobU persistently tended toward more frequent use, while WebU displayed more diversified use and lacked use or knowledge of several items.

In the area of healthcare, when informants were asked about search criteria and priorities for online health queries, a similar behavior pattern emerged. Table 1 shows that although consultation sources and priority of access to them were equal, MobU had higher consultation intensity and frequency.

### 3.4. Value and Use of Autonomous Community Websites

The first item to address in assessing the AC health services websites was how users accessed knowledge of their existence.

The great difference between the two profiles resided in how they knew about the existence of these websites. In the WebU group, 37.2% reported that this information was provided by the AC public health system, its personnel, or another administration (e.g., municipal), compared to 31.8% in the MobU group. For MobU, the most common way of accessing this knowledge was by internet search (38.6%), while for WebU, the percentage was lower (31.2%). This strikes a contrast between a more institutional and traditional access profile and a more informal and up-do-date one.

Turning the spotlight to frequency of access, we found that MobU accessed health websites considerably more often: 52.7% declared to have accessed such a site “in the last month” compared to 36.1% of WebU. The latter group also presented more sporadic access by declaring higher frequency of access superior to 3 months.

Levels of satisfaction with websites showed the greatest differences at “moderate” and “high” levels. Among WebU, 44.1% reported moderate satisfaction, compared to 36.4% of MobU. In contrast, 54.5% of MobU indicated “high” satisfaction, while only 48.4% of WebU declared the same. A general reading, thus, confirms a medium-high level of satisfaction in both profiles that is slightly higher among mobile users.

Concerning accomplishing the objectives for which they accessed the websites, there were no great differences: more than 9 out of 10 (92.4% of MobU and 90.8% of WebU) declared to have fulfilled the intended purpose of their visit.

The data compiled about the type of consultations on healthcare websites indicate that the information most commonly sought by both user groups was “locating medical centers, opening hours and contact information”. However, differences appeared for other types of information: MobU searched for “prevention programs” (vaccinations, advice on healthy habits …) and “Information on the AC public health system” more frequently than WebU. Differences for other types of information, such as “Illnesses, therapies and medications”, were smaller.

As for frequency, MobU used the websites more (there were fewer “never” responses) and more intensively (more frequent than occasional) than WebU.

For eHealth services, the pattern became more accentuated. First of all, the most frequent service accessed—by 83.3% of MobU and 72.5% of WebU—was online scheduling (changing, canceling) of appointments. The service most often accessed after that, though to a lesser extent (34.1% MobU, 28.0% WebU), was management of the medical identification card.

More specifically, differences found in the frequency of seeking online services indicate that 33% of MobU used the services mentioned on some occasion, compared to 28.8% of WebU. There was a difference of 10.9% between the two groups for online scheduling of appointments (16.7% of MobU had never used this service, compared to 27.5% of WebU). Apart from these two services, MobU reported using an average of 21.6% of available eHealth services (range 15–23%) and WebU used an average of 18% (range 13–20%).

Again, there were no great differences in the importance attributed to each eHealth service, although MobU tended to assign them more value and importance. There were fewer in that group who responded that these services had “No importance” (7.3% compared to 10.1% for WebU) and 66.2% of MobU considered all the services “very important”, compared to 63.2% of WebU.

Figure 2 shows the main differences among eHealth services and the prominence of services related to scheduling appointments and consulting/receiving medical reports or results.

Regarding ease of use of eHealth services, hardly any differences appeared, and both profiles considered them easy to use (88.3% of MobU; 88.1% of WebU). When asked for a general opinion of the AC websites, the opinions of both profiles were recorded along with MobU responses to a specific question about using those websites from a mobile device. Table 2 shows the comparison of opinions regarding AC public health websites by user profile.

To better understand the balance between categories, a visual indicator was created. Multipliers were used for each percentage of cases marked in the response categories (Very bad = *0; Bad = *1; Unremarkable = *2; Good = *3; Very good = *4). The maximum accumulated value of 400 points became a denominator that made it possible to standardize scores in a range of 0 (Very bad) to 1 (Very good). This indicator was applied to other questions in the same way, multiplying the scores by “*n* (number of categories)-1”.

As is apparent, there were no great differences between MobU and WebU ratings of AC websites, although the generally favorable opinions of MobU were somewhat attenuated when it came to evaluating the mobile versions of those websites.

Table 3 presents the results for other eHealth services based on the same calculation pattern.

For this, Table 3, Table 4 and Table 5, which have the same layout, the explanation of the colors is as follows: in columns WebU and MobU, the scores obtained are shown from lowest (red) to highest (green); intermediate values are shown in orange and yellow. In the Dif WebU-MobU columns, the differences in favor of MobU are highlighted in red; those features in which WebU scores higher are in green.

To summarize, use of eHealth services was generally lower among WebU than MobU and negative differences indicated an imbalance between the two user groups that favored the second. This was prominently the case for checking and receiving medical test results (with differences of 0.1632 and 0.1362 points, respectively) and checking waitlists (difference of 0.0753 points).

The scores were more moderate for the importance of the services: the user profiles were equal and gave significant relevance to a good portion of the services. Again, MobU assigned greater importance to receiving medical results, the possibility of sending images or files, and online consultations by videoconferencing (0.0627 points, 0.043 points, and 0.042 points difference, respectively, compared to WebU).

Following the same scoring protocol, we explored user expectations regarding the potential of eHealth services by proposing several items to evaluate the degree of support or help that ICTs might lend to this area. The results are displayed in Figure 3.

Generally, ICTs were considered a very helpful support to Health services, as the expectation was greater than 0.7 points out of one. Remarkably, both profiles gave great importance to facilitating medical administrative tasks online, avoiding unnecessary trips, and being better informed. In total, 9 out of 10 users favored these uses.

Though the results for both groups were again fairly equal, MobU placed greater emphasis on the potential of ICTs, with especially noteworthy differences in the response concerning “more direct and immediate interaction with healthcare professionals”. This may be related to MobU’s appreciation of videoconferencing with medical professionals (which was highlighted in the previous graph) and easier administrative processing.

The results on the opinions about what actions might allow more people to use eHealth tools can be seen in Table 4.

All the proposed options were highly rated, with several initiatives scoring above 0.9 out of 1 (maximum importance): improving coordination between medical facilities and personnel, simplifying online administrative tasks, increasing data security and privacy, improving citizen ICT skills, improving website and app usability, and expanding coverage and access nationally. One difference between MobU and WebU appeared regarding the economic factor, with the latter assigning greater importance to lowering internet connection costs.

According to the scores for the changes that ICTs have generated in the current public health system, 77.7% of MobU and 76.2% of WebU confirmed that new technologies have changed the relationship between users and health services. When asked about the direction of these changes, 97.03% of MobU and 91.98% of WebU rated them positively, though the latter were more doubtful (4.76% rated the changes as “Neither positive nor negative” and the remainder responded “Negative”).

Those who considered the changes positive or negative were asked how ICTs improved health services and what needed to happen for ICT-related changes to be positive.

For the first question (how ICTs improved health services), several options were provided and the interviewees were asked to choose a maximum of two. The results in Figure 4 indicate that both profiles coincided in their priorities: convenience, speed, ease of access, ease of communication, and economy. Only on the last of these did MobU and WebU show differences; in general, MobU emphasized the advantages more (5% difference over WebU) in the categories of ease of access, speed, and economy (4.79%, 4.53%, and 4.41%, respectively).

For the second question, regarding what could be done to make ICT-related changes to health services positive, a long list of items was prepared. The most prominent—those indicated by at least 3% of the sample or that showed at least 1% difference between the two user profiles—are listed in Table 5.

In consonance with earlier opinions in which WebU was less optimistic or less favorable to ICTs, here they were the group that indicated most actions needed. In fact, the greatest variation between the profiles was detected in the level of satisfaction expressed by MobU: almost 25% indicated that no specific action was needed (“Nothing/nothing in particular/it’s already good”) compared to 16.70% of WebU.

User responses indicated that the most prominent measures that could improve the impact of ICTs on health services were: “More information/have more information available/access to all fields (medications, prevention …)”; “Speed/Rapid service/diagnosis, consultations/Rapid attention”; “Immediate contact/more direct/direct communication with doctor/medical professionals/health center”. All of these had values of around 10% (or even higher, as can be observed) of users in both profiles. Following those were improvements such as “Convenience in doing administrative procedures/Save trips” and “Simplify administrative procedures/make procedures easier”, with percentages that varied from 7–9%.

The data in the table indicate that more WebU believed that there is undeveloped ICT potential in relation to health care.

### 3.5. Sociodemographic Context

The results indicated that 51.5% of the women surveyed accessed health services using their mobile phones, compared to 48.5% of men. For their part, 52% of men accessed these services through a website, compared to 48.1% of women.

Habitat was an important sociodemographic profile factor for both user types. The data reveal significant differences: 37.9% of MobU lived in municipalities of fewer than 25,000 inhabitants, which is a higher percentage than WebU (33%). There were scarcely any differences in the range of 25,000 to 100,000 inhabitants, and there were more WebU (39.4%) than MobU (34.1%) in large cities of more than 100,000 inhabitants.

Most MobU (47.7%) were in the 25–39 age range, followed by 33.3% in the 40–65 range. The inverse was found for WebU, most of whom were in the 40–65 age range (48.8%), followed by 36% in the 25–39 range. The difference between profiles was also noteworthy in the 18–24 age range, which comprised almost 16% of MobU compared to 10.6% of WebU.

The most frequent educational level for MobU (42.7%) corresponded to upper secondary (advanced vocational, university track baccalaureate), followed by university (19.8%). The pattern was replicated for WebU (37% and 26.6%, respectively), though university studies had greater weight.

The majority in both groups declared that they were employees (56.8% MobU and 53.2% WebU). Occupation data were also noteworthy: 11.2% of WebU were retired or pensioners, and there were more homemakers in MobU (3.8%) than WebU (1.8%).

The most common MobU family profile was composed of four members (30.8%), followed by three (27.7%) and two (23.1%) members. In WebU, families composed of two and four individuals were co-dominant (28.1%), followed by families with three members (26.3%).

Less than 5% of both profiles have incomes of less than 600 euros per month (MobU, 2.5%; WebU 4.4); in a middle-income level (601 to 1800 euros), there are 49.2% MobU and 44.4% WebU; finally, in the highest income level (more than 1800 euros), there are 48.4% MobU and 51.2% WebU. Therefore, both profiles are located on a medium-high social level in terms of income. That contrasts with the overall situation of Spain, which is not so favorable.

## 4. Discussion

The results confirm two sociodemographic user profiles with several differences. The main MobU profile is that of a woman between 25 and 39 years of age with upper-secondary education who works for an employer and lives in small towns or rural settings (less than 25,000 inhabitants) with a family composed of four members and a net monthly income of 1801 to 2500 euros.

Meanwhile, the predominant WebU profile is that of a man, 40–65 years of age, with upper secondary or university education, who works for an employer, lives in large urban settings in a family of two to four members and has a net monthly income similar to that of MobU (1801 to 2500 euros).

Other data sources on internet connection in Spain [43,44,45] were used to compare general user profiles along with those of eHealth (WebU) and mHealth (MobU) users. The results brought some interesting considerations to light.

The results for sex largely coincided with those of the ONTSI studies to confirm that women have a higher daily connection frequency than men: 70.3% of the women surveyed declared that they connected to the internet more than ten times per day, compared to 60.7% of men [44]. When the data was contextualized, the mobile phone constituted the main point of internet access for Spanish female users (99.8%).

Similarly, INE data [45] indicates that women access the internet more habitually (at least 5 days per week) than men (83.8% and 82.4%, respectively) and use eHealth services more often. Women also accessed personal health files (17.8%) and online health services (22.1%) more than men (17.3% and 18.9%, respectively); they also scheduled more medical appointments using the internet (43.6% vs. 36.9%) and looked for more health-related information (73% vs. 61.1%).

In the same way, the ONTSI data for age shows differential behavior that is inversely related to connection frequency. The youngest age group accessed the internet from a mobile phone most frequently: 80.7% of people aged 15 to 25 connected to the internet more than 10 times daily, compared to only 42.2% of those over 65 years of age [44]. These findings converge with those of the present study (eHealth Survey), which situates MobU as the youngest and most active profile on the internet generally and in the field of eHealth specifically.

Educational level showed some significant differences, to the point that 7 out of every 10 people with primary education connected to the internet weekly. The percentage was practically 100% among people with university studies. Meanwhile, 27% of those who used the internet weekly had university studies and 15% had basic studies [43].

In the WebU and MobU profiles, the reality was similar with regard to case distribution; people with basic education used eHealth services much less. There were no great differences in use after the threshold of secondary education, the category that represented most people in both profiles. Though educational level may be relevant, it did not determine access to eHealth and mHealth for these profiles. However, it would be an impediment for people who are not currently users.

Finally, some differences in the use of technological devices came to light by looking at income level. Those who lived in a home with a high income level (more than 3000 euros monthly) declared that they used or have used all types of devices to a larger degree than the other groups. This situation diverges from that of Spanish society as a whole, with higher levels of poverty, which highlights that these tools do not allow for greater social inclusion or generating equity, but rather the opposite. However, the idea of the mobile phone as a reference device is supported if we look at frequency of use: almost all those interviewed used a smartphone daily and habitually used it several times per day [44].

Apart from the sociodemographic profile, the results generally confirm what the literature indicates about the priorities of the collectives (perspectives of service providers and users), but also reveal shared concerns that might seem less obvious at first.

Again, ease of use and simplicity of services and websites were requirements that users shared with medical professionals, as indicated in previous studies [24]. However, elements located on a secondary plane of user priorities in other research shifted to the forefront [25,26]. Specifically, data security and reliability (especially in the section on privacy) emerged as concerns for both MobU and WebU. This area has gained relevance with the boom of mobile apps—to be discussed shortly—which may explain its importance to both groups.

Regarding expectations about mHealth, MobU were more critical of the mobile versions, though they generally had a better opinion of eHealth services. This may imply that expectations were similar but the services were not well-adapted, which reduced satisfaction with mobile formats. Thus, the problem is about usability: an element that involves users as well as medical professionals, albeit from different perspectives [25,26,27].

Finally, several studies indicate that since the outbreak of the pandemic, eHealth and mHealth have positioned themselves as positive initiatives for reducing healthcare system overload and pressure on hospitals [38,39]. However, the elements and capabilities needed for health services to benefit from the potential of ICTs are not yet established. Such an undertaking requires flexible financing agreements, adequate training of medical personnel, significant changes in management, and the redesign of existing patient care models [35,40]. From the results, we can deduce that eHealth services in Spain are not predominant among users. Weaknesses were also detected in knowledge and use of eHealth services in the general population as well as in the usability and development of mobile versions, a format with greater projection and social scale [43,44,45].

One important aspect to keep in mind when reading these data and results is that the National Health System in Spain includes 17 separate autonomic systems, as described in the introduction. This decentralization entails distinct eHealth services and levels of provision, which explains part of the variability in the results. However, a wide range of services have already been implemented and this reality is not beyond the scope of research. In fact, the survey response items addressed questions or elements that any inhabitant anywhere in Spain could evaluate, regardless of regional specificities associated with access to healthcare services.

## 5. Conclusions

The first conclusion that can be extracted from this study is that both MobU and WebU had a very similar understanding of eHealth. Essentially, they shared the same expectations, opinions, and appreciation of eHealth.

However, more differences appeared in sociodemographic profiles. The literature on accessibility [28,29] has highlighted the mobile phone as the main element for contributing to the achievement of healthcare equality, due to its portability, price, and use and connection cost. The use of mHealth will probably continue to grow and develop in step with new connectivity technologies that facilitate it. In fact, the expansion of mHealth could even help “socialize” mobile technology [28,29]. For both chronic and occasional users, mobile phones may help activate patients to seek information and take greater care of their health, giving rise to what is known as the ePatient [33].

From the results, we can conclude that people who use their mobile device as an access point are younger (predominantly aged 25 to 39) and have a similar (or slightly better) health status but visit the doctor more frequently than users who access eHealth from other devices.

This information may induce ideas of greater concern for health, and subsequent monitoring, as well as more online interaction with medical administrations. In any case, we can highlight that MobU generally had a more positive outlook and greater sense of the potential and capacity of ICTs than WebU, though both profiles considered the existence and promotion of online health services to be important going forward. According to the results, WebU attitudes were more expectant than convinced, while MobU were more cyber-optimists [46], as reflected in their scores for satisfaction with services.

Though the hypothesis would necessarily need to be verified, the data on internet use, places of connection, and type of devices used indicate that MobU are more concerned or aware of health issues. Important differences were also detected in how much they searched for health information online compared to WebU.

As expected, MobU clearly had a more digital profile than WebU for general internet habits, including the sphere of healthcare. MobU had knowledge of eHealth services/information through their own initiative—online searches—while more factors came into play for WebU (e.g., peer groups and administrations) and more institutional pathways that were not necessarily linked to ICTs.

Most MobU had visited their AC health website at least once in the last year (WebU were much more sporadic in their access). MobU searched for more—and more varied—information and services than WebU.

The same occurred for requesting services online; the results confirmed that MobU had more knowledge of available services and used them more than WebU.

Regarding potential, MobU emphasized the advantages of eHealth and mHealth services and ICTs in general—such as saving trips, time, and resources—as well as ease of access. This could fit with their habitat profile, as MobU tended to live more in rural settings or small towns (fewer than 25,000 inhabitants).

Despite this, the more positive assessment of MobU does not seem attributable to the design of mobile versions of the AC websites. Both MobU and WebU considered the habitual websites to be better that the mobile versions.

Mobile apps are penetrating AC health systems, generating a new ecosystem of health applications designed integrally for mobile formats (Android or iOS) that did not exist at the time of this analysis. These apps began to develop in Spain in late 2018, and expanded to all ACs in 2020, under the unexpected and vigorous impulse of the COVID-19 global health crisis. The strong expansion, use, and foreseeable development of these technologies has established mHealth as a dimension of eHealth and made it an object of study from various focal points and perspectives [35].

At the beginning of this new era, the research presented here constitutes a data source for comparing progress and verifying the development that users have been experiencing since 2020. It provides an opportunity to reflect on these changes and evaluate their impact on Spanish public health.

## Figures and Tables

**Figure 1 ijerph-18-13055-f001:**
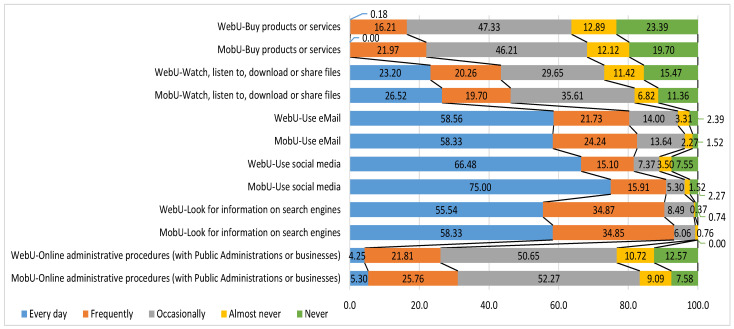
Basic online activities (by user profile).

**Figure 2 ijerph-18-13055-f002:**
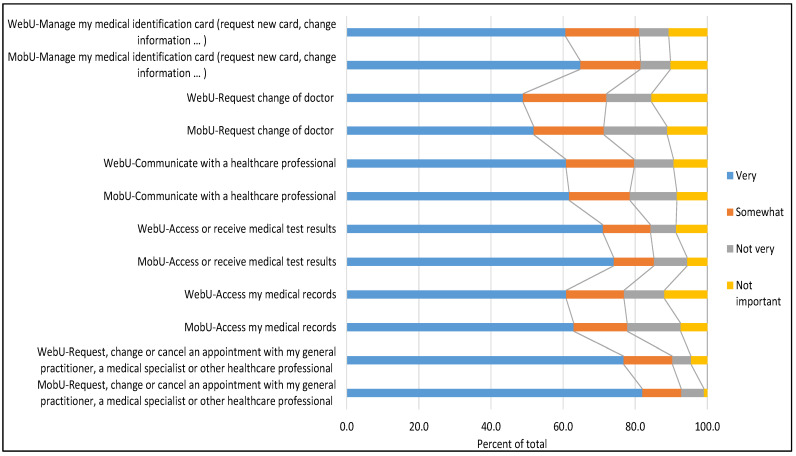
Ranking of the importance of health services (by user profile).

**Figure 3 ijerph-18-13055-f003:**
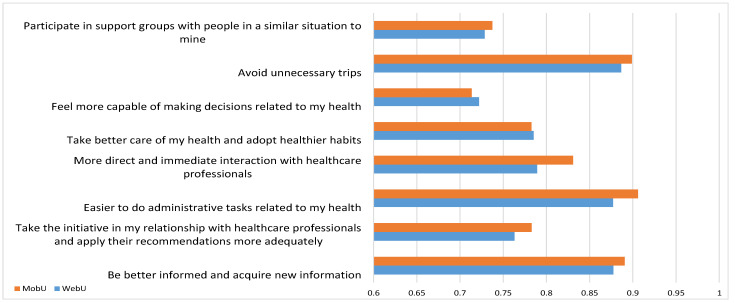
Rating of the potential of new technologies in health-related actions (by user profile).

**Figure 4 ijerph-18-13055-f004:**
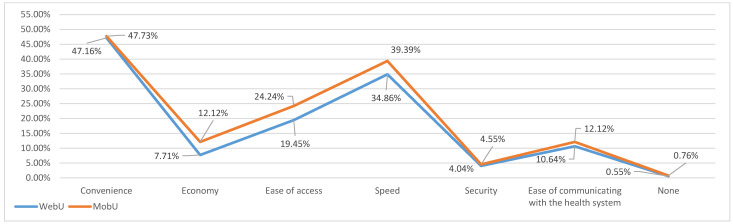
Advantages of ICTs for health services (by user profile).

**Table 1 ijerph-18-13055-t001:** Frequency of accessing health information sources (by user profile).

When You Look For Information about Health, How Often to You Go to the Following Websites?										
Frequency of Access	Every Day		Frequently		Occasionally		Almost Never		Never	
User Profile	WebU	MobU	WebU	MobU	WebU	MobU	WebU	MobU	WebU	MobU
My Autonomous Community’s public health service	0.92	0.00	11.58	18.94	43.20	39.39	19.49	18.18	23.48	24.82
Another institution such as the World Health Organization or the Ministry of Health	0.18	0.00	3.31	6.82	19.12	18.18	17.65	18.94	56.06	59.74
Businesses that provide healthcare services (insurance companies or workplace mutuals)	0.00	0.00	2.94	3.03	17.28	17.42	14.89	12.88	66.67	64.89
Patient associations	0.18	0.00	2.21	3.82	11.23	6.87	15.47	16.79	72.52	70.90
Health blogs or forums	0.37	0.76	4.97	8.40	23.57	23.66	19.71	19.08	48.09	51.38

Source: authors’ own data. Highest and lowest frequencies/scores are showed in different colors.

**Table 2 ijerph-18-13055-t002:** Opinions of Autonomous Community health websites (by user profile).

	Frequency			Score		
	Rating of AC Websites		Rating of Mobile Version of AC Websites	Rating of AC Websites		Rating of Mobile Version of AC Websites
	WebU	MobU	MobU	WebU	MobU	MobU
Very bad	0.4	0.8	3.1	0	0	0
Bad	2.5	3.9	1.5	2.46	3.88	1.53
Unremarkable	19.3	15.5	22.9	38.64	31.01	45.80
Good	66.7	65.9	55.7	200.00	197.67	167.18
Very good	11.2	14.0	16.8	44.70	55.81	67.18
Score (0–1)				0.7145	0.7209	0.7042

Source: authors’ own data.

**Table 3 ijerph-18-13055-t003:** Frequency and importance of eHealth services (by user profile).

	Scores Regarding Frequency of Use of eHealth Services			Scores Regarding Importance of eHealth Services		
	WebU	MobU	Dif WebU-MobU	WebU	MobU	Dif WebU-MobU
Request a second medical opinion	0.0523	0.0811	−0.0288	0.7509	0.7519	−0.0010
Register a complaint	0.0607	0.0848	−0.0241	0.7857	0.8010	−0.0154
Access medical test results	0.1444	0.3077	−0.1632	0.7563	0.7667	−0.0104
Receive medical test results	0.1202	0.2564	−0.1362	0.7706	0.8333	−0.0627
Check an appointment or position on a waiting list	0.1948	0.2701	−0.0753	0.8362	0.8640	−0.0278
Check prescriptions or medications	0.1236	0.1411	−0.0176	0.7259	0.7275	−0.0016
Request reimbursement of expenses	0.0486	0.0751	−0.0265	0.7031	0.7025	0.0007
Receive information on preventative health campaigns	0.1118	0.1042	0.0077	0.7519	0.7743	−0.0224
Consult with a healthcare professional online by videoconference	0.0117	0.0061	0.0056	0.7392	0.7813	−0.0420
Monitor health using remote measuring devices (glucose, blood pressure …)	0.0760	0.0841	−0.0081	0.7500	0.7787	−0.0287
Compile data about individual physical activity (smartwatches, pulsometers)	0.1566	0.1976	−0.0411	0.6509	0.6432	0.0077
Ability to send health-related photos or files to a professional	0.0513	0.0685	−0.0171	0.7012	0.7442	−0.0430
Average	0.0960	0.1397	−0.0437	0.7435	0.7640	−0.0206

Source: authors’ own data. Highest and lowest frequencies/scores are showed in different colors.

**Table 4 ijerph-18-13055-t004:** Importance of measures to foster access to eHealth (by user profile).

	WebU	MobU	Dif WebU-MobU
Improve internet speed	0.8579	0.8590	−0.0011
Improve coverage and access in all of Spain	0.9002	0.9033	−0.0031
Reduce internet costs	0.8956	0.9192	−0.0236
Reduce costs of devices (mobile phones, computers, tablets)	0.8625	0.8687	−0.0062
Improve citizen education in new technologies	0.9233	0.9116	0.0116
Increase security and privacy of personal data	0.9331	0.9338	−0.0008
Make known and simplify the use of digital certificates and national ID cards	0.8887	0.9000	−0.0113
More encouragement from medical professionals to patients to make greater use of these technologies	0.8278	0.8244	0.0034
Publicize and inform more about health websites and apps	0.8520	0.8564	−0.0044
Better coordination among medical centers, hospitals, pharmacies and professionals	0.9501	0.9495	0.0006
Simplify administrative processing of services offered	0.9425	0.9389	0.0036
Increase the range of services	0.8810	0.8821	−0.0011
Make better use of websites and apps	0.9012	0.9128	−0.0116
Average	0.8935	0.8969	−0.0034

Source: authors’ own data. Highest and lowest frequencies/scores are showed in different colors.

**Table 5 ijerph-18-13055-t005:** Actions that could improve the impact of ICTs on health services (by user profile).

	WebU	MobU	Dif WebU-MobU
Speed/Rapid service/diagnosis, consultations/Rapid attention	10.64%	7.58%	3.07%
Agility, agility in service/administrative procedures	4.22%	2.27%	1.95%
Immediate contact/more direct/direct communication with doctor/medical professionals/health center	9.17%	9.09%	0.08%
Simplify administrative procedures/make procedures easier	8.26%	8.33%	−0.08%
Reduce/speed up waiting lists/check waiting lists	4.59%	6.06%	−1.47%
More information/have more information available/access to all fields (medications, prevention …)	11.56%	9.09%	2.47%
Convenience in doing administrative tasks and procedures/Save trips	9.17%	7.58%	1.60%
Have access to test results	2.94%	3.03%	−0.09%
Have access to medical records/history	3.85%	0.76%	3.10%
Easy access to internet/easy to use/easy/accessible/fast/interactive /app	4.77%	4.55%	0.23%
Save time	1.10%	0.00%	1.10%
Improve customer service	1.47%	0.00%	1.47%
Better coordination of visits/tests/doctors (all coordinated)	2.75%	1.52%	1.24%
Cannot comment because does not use/not interested in new technologies/prefers face-to-face	4.95%	3.03%	1.92%
Nothing/nothing in particular/it’s already good	16.70%	23.48%	−6.79%

Source: authors’ own data. Highest and lowest frequencies/scores are showed in different colors.

## Data Availability

The survey will be available in this public repository: http://www.arces.cis.es/jEstudios.jsp.

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
