# Peer review of "eHealth and mHealth Development in Spain: Promise or Reality?"

_ijerph, 2021, doi:10.3390/ijerph182413055_

Round 1
Reviewer 1 Report
The paper is an investigation into ehealth and mhealth within Spain.
Overall the paper is an interesting read and, given the increasing use of mhealth, a potentially important addition to this area of investigation.
The paper would benefit from:
- A further edit for English
- Greater detail related to the methods - how was the data actually analysed.
- The earlier paragraphs use a space between each, the latter an indent. This should be consistent.
- Lines 199-201 appear to need to be removed.
- The figures appear to have two titles (one at the top and one at the bottom) which is confusing.
- It would be useful to include numbers and not just percentages in the tables.
- This paper is descriptive in nature, but focuses upon webu versus mobs. It might add to the paper to see if there are statistically significant differences between the two groups. At present, there is no analysis of whether the differences by sociodemographic status etc are statistically significant. For example, it might be interesting to see if the differences by education were meaningful. At present, two groups are investigated without any statistical analysis of potential differences.
Reviewer 2 Report
The report is well written and adequately presented. Please see my minor comments as stated below:
- Please, consider moving some details of the results as an annex with supplementary material. This would easy the reader to focus on the main important results
- Please, the reader would benefit if the original survey would be made available, as an annex or with a link.
- Please, the reader would benefit from a shortened version of the conclusions. Some of the current conclusions could be better located in a summary of findings section.
- The caption of some tables would benefit from additional description. This is the case, for example, of tables 3, 4 and 5, in which the meaning of the different colours might be explained.
